# Pan-neuronal screening in *Caenorhabditis elegans* reveals asymmetric dynamics of AWC neurons is critical for thermal avoidance behavior

Ippei Kotera[1], Nhat Anh Tran[1], Donald Fu[1], Jimmy HJ Kim[2], Jarlath Byrne Rodgers[1,3], William S Ryu[1,2]*

[1]Donnelly Centre, University of Toronto, Toronto, Canada; [2]Department of Physics, University of Toronto, Toronto, Canada; [3]Department of Cell and Systems Biology, University of Toronto, Toronto, Canada

**Abstract** Understanding neural functions inevitably involves arguments traversing multiple levels of hierarchy in biological systems. However, finding new components or mechanisms of such systems is extremely time-consuming due to the low efficiency of currently available functional screening techniques. To overcome such obstacles, we utilize pan-neuronal calcium imaging to broadly screen the activity of the *C. elegans* nervous system in response to thermal stimuli. A single pass of the screening procedure can identify much of the previously reported thermosensory circuitry as well as identify several unreported thermosensory neurons. Among the newly discovered neural functions, we investigated in detail the role of the AWC$^{OFF}$ neuron in thermal nociception. Combining functional calcium imaging and behavioral assays, we show that AWC$^{OFF}$ is essential for avoidance behavior following noxious heat stimulation by modifying the forward-to-reversal behavioral transition rate. We also show that the AWC$^{OFF}$ signals adapt to repeated noxious thermal stimuli and quantify the corresponding behavioral adaptation.

*For correspondence: willryu@gmail.com

**Competing interests:** The authors declare that no competing interests exist.

## Introduction

*C. elegans* is one of the simplest multicellular organisms with only 302 neurons in hermaphrodites. Our current understanding of its neural functions involves three major levels of hierarchical systems: genes, neurons, and behaviors. In the neuroscience of *C. elegans*, the most common approach to study neural functions is to apply a perturbation at the genetic or cellular level through techniques such as mutation, cell-ablation, or optogenetic stimulation, and then screen the worm at the behavioral level. Although proven to be effective, this approach is far from ideal because the behavioral screening is performed in a different hierarchical level than the level at which the perturbation was initially applied (*Bhalla and Iyengar, 1999*). The general difficulty in making connections between these different hierarchies has hindered our ability to elucidate neural mechanisms connecting genes to behavior in *C. elegans* despite its extreme simplicity.

It would be advantageous if we could directly screen for the neural functions in *C. elegans*, because then we could apply perturbations and interpret the results all within one hierarchical level. Through the power of optogenetics, it has become possible to make parallel measurements of multiple neurons in *C. elegans* via 3D imaging of calcium activities (*Prevedel et al., 2014*; *Schrödel et al., 2013*; *Kato et al., 2015*). More recent work describes methods to record pan-neuronal activities in moving worms using spinning disk confocal microscopy (*Venkatachalam et al., 2016*). Here we have taken an integrated approach and developed a pan-neuronal functional

screening system in which the neural activities from many of the neurons are recorded concurrently in an intact animal in response to precisely applied thermal stimuli, similar to previously reported systems with some customization to meet our experimental needs. We then connected these signals to behavioral outputs by repeating the thermal stimuli in carefully quantified behavioral assays.

Using pan-neuronal screening, we have discovered that the calcium flux in a pair of AWC neurons respond in opposite directions in reaction to noxious heat stimulation. AWCs are the amphid sensory neurons located in the head region of *C. elegans*, and are crucial for the sensing of certain volatile chemicals. Unlike most other pair-wise neurons, AWC neurons are functionally asymmetric: the AWC$^{ON}$ neuron senses butanone odor, and AWC$^{OFF}$ senses 2,3-pentanedione, while both neurons symmetrically sense benzaldehyde and isoamyl alcohol (*Wes and Bargmann, 2001*; *Chalasani et al., 2007*), which are essential for proper chemotaxis (*Clark et al., 2006*; *Mori and Ohshima, 1995*).

In thermosensation, AFD is the primary sensory neuron and is essential for thermotaxis (*Mori and Ohshima, 1995*). As for the sensory neuron AWC, there have been some contradicting reports in terms of its involvement in thermosensation and thermotaxis. One group reports that AWC exhibits AFD-like continuous calcium transients in temperature ramps, and that its thresholds depend on the previously exposed temperature (*Kuhara et al., 2008*). Another group also reports AWC's role in thermosensation but their calcium transients are more similar to interneuron AIY signaling (Figure 2B), in the way that they consist of short stochastic transients whose frequency responds to the previous temperature (*Biron et al., 2008*). Also there is an electrophysiological study which detected no membrane current in AWC neurons in the thermal ramps (*Ramot et al., 2008*). A more recent study states that removal of AWC by laser ablation or cell-specific recCaspase expression results in no significant disruption either in positive or negative thermotaxis (*Luo et al., 2014*).

In our pan-neuronal study, AWC's signal in response to temperature ramps shown prior to induce thermotactic responses was small and difficult to measure. However, when we applied either a very fast temperature rise (a few degrees in less than 50 ms) or high absolute temperature (33℃), we detected very deterministic AWC signals that are asymmetric in AWC$^{ON}$ and AWC$^{OFF}$: the calcium transients in AWC$^{OFF}$ neurons are always positively correlated to the nociceptive thermal stimuli whereas the transients in AWC$^{ON}$ neurons are negatively correlated. This novel activity was then tested in freely moving animals in which the nociceptive stimulation was applied to further investigate their role in physiological behaviors utilizing asymmetry mutants and laser ablation of AWCs. Indeed, the thermal avoidance behavior, which has been linked to the activities in the AFD, FLP, PVD, and PHC neurons (*Chatzigeorgiou et al., 2010*; *Liu et al., 2012*; *Mohammadi et al., 2013*) is also strongly coupled to the AWC$^{OFF}$ neurons but not AWC$^{ON}$. By combining pan-neuronal calcium imaging with our novel behavioral analyses, we found that the AWC$^{OFF}$ neuron is essential and sufficient for noxious heat sensation and the subsequent avoidance behavior.

## Results

### Pan-neuronal calcium imaging coupled with thermal perturbations reveals novel neural functions

In order to investigate the calcium dynamics in the nervous system of *C. elegans*, we have developed an imaging system with fast z-scan, multicolor, and thermal stimulation capabilities, along with a software package for automated system operation, image registration, cellular segmentation, gradient vector field (GVF)-based cell tracking (*Li et al., 2007*), and GPU-based 3D deconvolution (*Bruce and Butte, 2013*). The system utilizes wide-field illumination with post-acquisition deconvolution rather than confocal optics, allowing for very efficient collection of fluorescence in multiple optical planes. This combination has enabled fast acquisition of pan-neuronal images (20 fps) with minimal photo-toxicity and photo-bleaching (up to 45 min acquisition, *Figure 3—figure supplement 2*) using conventional calcium indicators expressed in the neurons of *C. elegans*.

We have generated transgenic *C. elegans* lines expressing a genetically encoded calcium indicator (*Zhao et al., 2011*) (G-GECO 1.1 coupled with DsRed2) in the nuclei of all the neurons for ratiometric pan-neuronal calcium imaging (*Figure 1A*). As previously reported (*Schrödel et al., 2013*), we used nuclear-targeted indicators because the small size and compactness of the *C. elegans* nervous system makes whole neuron segmentation very difficult. Due to some motion artifacts, the ratiometric indicator was critical for the stable measurement of neural activities. We also co-

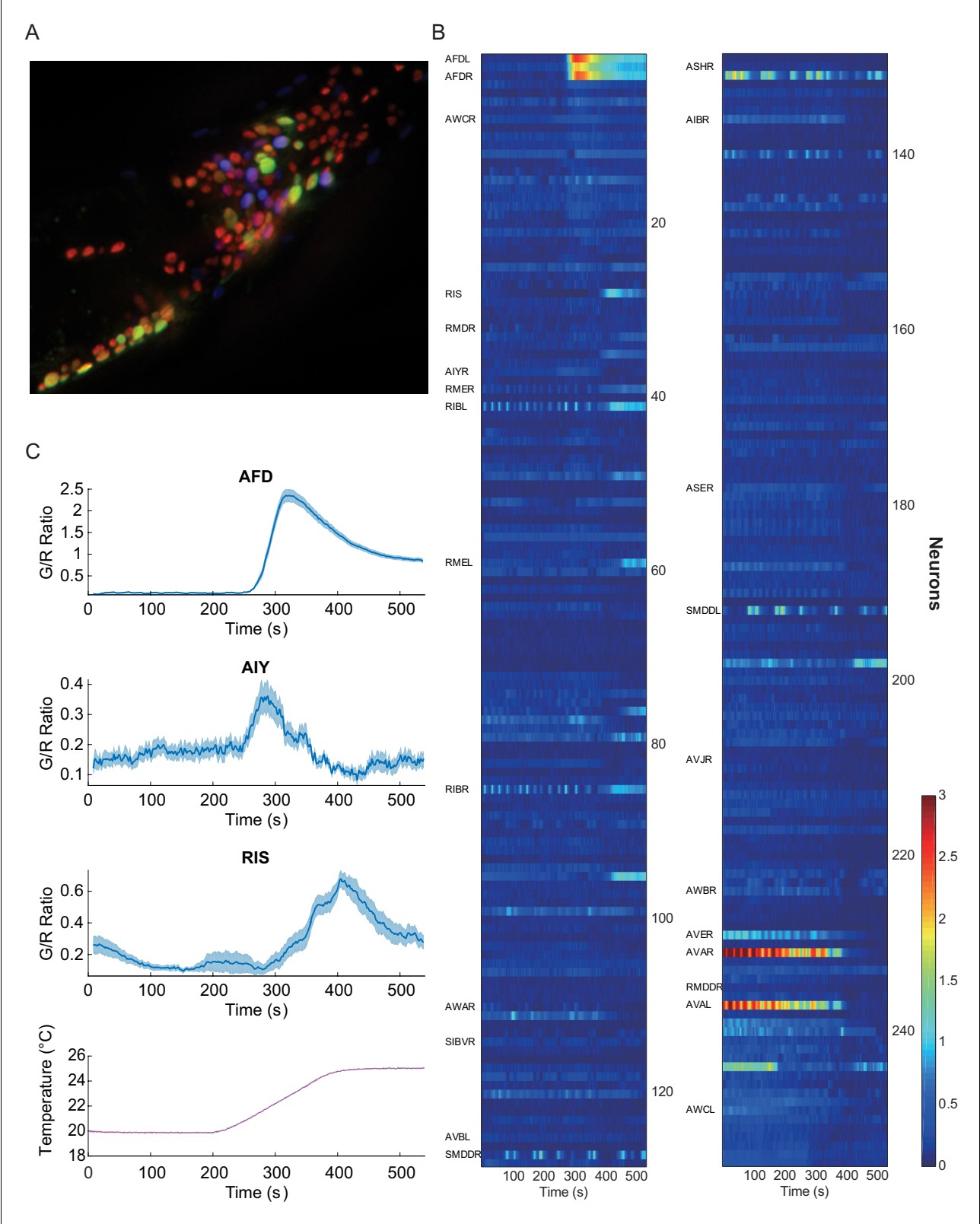

**Figure 1.** Pan-neuronal calcium signals in response to thermal ramp. (**A**) A maximum-projection image of nuclear fluorescence in the head region of *C. elegans*. Pan-neuronally expressed G-GECO1.1 and DsRed2 are psuedocolored in green and red, respectively. *glr-1p::mNeptune* (psuedocolored in blue), among other markers, was used to help identify some of the head neurons. (**B**) A heat-map representation of the whole-brain calcium transients. Calcium activities are shown in color: the larger indicator ratio is expressed as reddish color while smaller is in blue. Each row is a calcium recording

*Figure 1 continued on next page*

*Figure 1 continued*

from a single neuron, and the time-series are sorted by correlation coefficient to AFD activity. Neurons identified for this study are labeled on the left side. (C) Thermotactic calcium responses during a temperature ramp (bottom) in AFD (top, *n* = 38, see also *Figure 1—figure supplement 1*), AIY (second panel, *n* = 14), and RIS (third panel, *n* = 18) neurons in adult worms which had been cultivated at 23°C. The intensity of the calcium indicator (G-GECO1.1) was divided by the intensity of the bicistronically coexpressed reference (DsRed2), and the ratios (G/R ratio) were plotted as a function of time. All measurements were made in the nuclei. Note that y-scale is different among the neurons. Error bars indicate standard errors (shaded areas in light blue). Source data are available in Figure_1-source_data_1.mat.

The following figure supplements are available for figure 1:

**Figure supplement 1.** $T_C$-dependent thermosensory calcium responses in the nuclei of AFD neurons, extracted from the pan-neuronal recordings.

**Figure supplement 2.** Some of the representative calcium transients in the pan-neuronal measurements in response to a temperature ramp (same as *Figure 1—figure supplement 1*).

**Figure supplement 3.** Cell identification by glr-1p::mNeptune marker.

---

expressed various cell specific neuronal markers (such as *glr-1p::mNeptune*, *tax-4p::mNeptune*, *odr-2p::mNeptune*) to accurately identify neurons in the head region.

Using this imaging strategy (see Materials and methods), we detected strong calcium signals in the nuclei of the command, motor, and sensory neurons (*Figure 1B*), but not in the nuclei of some interneurons such as RIA, which is in agreement with previous reports where the signals were detected in the neurites and cell body, but not in the nucleus (*Hendricks et al., 2012*). Despite this shortcoming, we have successfully measured neural activities in many neurons under various thermal conditions and perturbations.

As a proof of concept of this approach, we first screened the neuronal activities for a well-studied sensory response—thermotaxis. As previously reported (*Clark et al., 2006*), we detected very strong and deterministic signals in the nuclei of AFD thermosensory neurons, when we gradually increased the worm's temperature, consistent with a dependence on the cultivation temperature ($T_C$) (*Figure 1C*, top, and *Figure 1—figure supplement 1*). We also detected similar but smaller signals from the nuclei of AIY interneurons (*Figure 1C*, second), as shown previously, confirming the efficacy of this approach (*Clark et al., 2006*). We then screened for temperature-dependent signal changes in previously unreported neurons. One notable neuron was the GABAergic RIS neuron, which previously has been linked to quiescence but not thermosensation (*Turek et al., 2013*). Unlike AFD or AIY neurons, the signals from the RIS neuron are broad and less consistent, but they are nonetheless dependent on the previously exposed temperature on average, and start to rise about 3°C above the $T_C$ (*Figure 1C*, third). RIS is one example in this screening, and there were other neurons such as RMDV and SMDV, which showed negative correlation to the AFD or RIS activity (*Figure 1B* and *Figure 1—figure supplement 2*). These results demonstrate the effectiveness of the pan-neuronal functional screen in finding novel neural functions in response to external perturbations.

## AWC neurons respond asymmetrically to noxious thermal stimuli

Thermal nociception and thermotaxis involve mostly separate signaling pathways in *C. elegans* (*Glauser et al., 2011*; *Wittenburg and Baumeister, 1999*) but the circuitry of thermal nociception is not as well understood, so we next sought to identify novel neural functions in response to noxious thermal stimuli. Instead of a thermoelectric heat pump, we used a focused infrared laser (1440 nm) to rapidly heat up the head region of *C. elegans*. The laser beam was carefully controlled so that the local temperature rises from 23°C to 33°C within a few tens of milliseconds and held for 20 s with an interstimulus interval (ISI) of 30 s.

Similar to our thermosensory measurements, both the left and right AFD neurons showed the most distinct calcium signals with strong positive correlation to the noxious thermal stimuli (*Figure 2A*). Other sensory neurons such as FLP and thermotactic interneurons such as AIY also had positive correlation, albeit with reduced signal strength (*Figure 2B*). In contrast to the thermal ramp stimulation, the noxious thermal stimuli did not consistently invoke signals in the aforementioned RIS, RMDV, and SMDV neurons. Interestingly, AWC sensory neurons produced asymmetric calcium

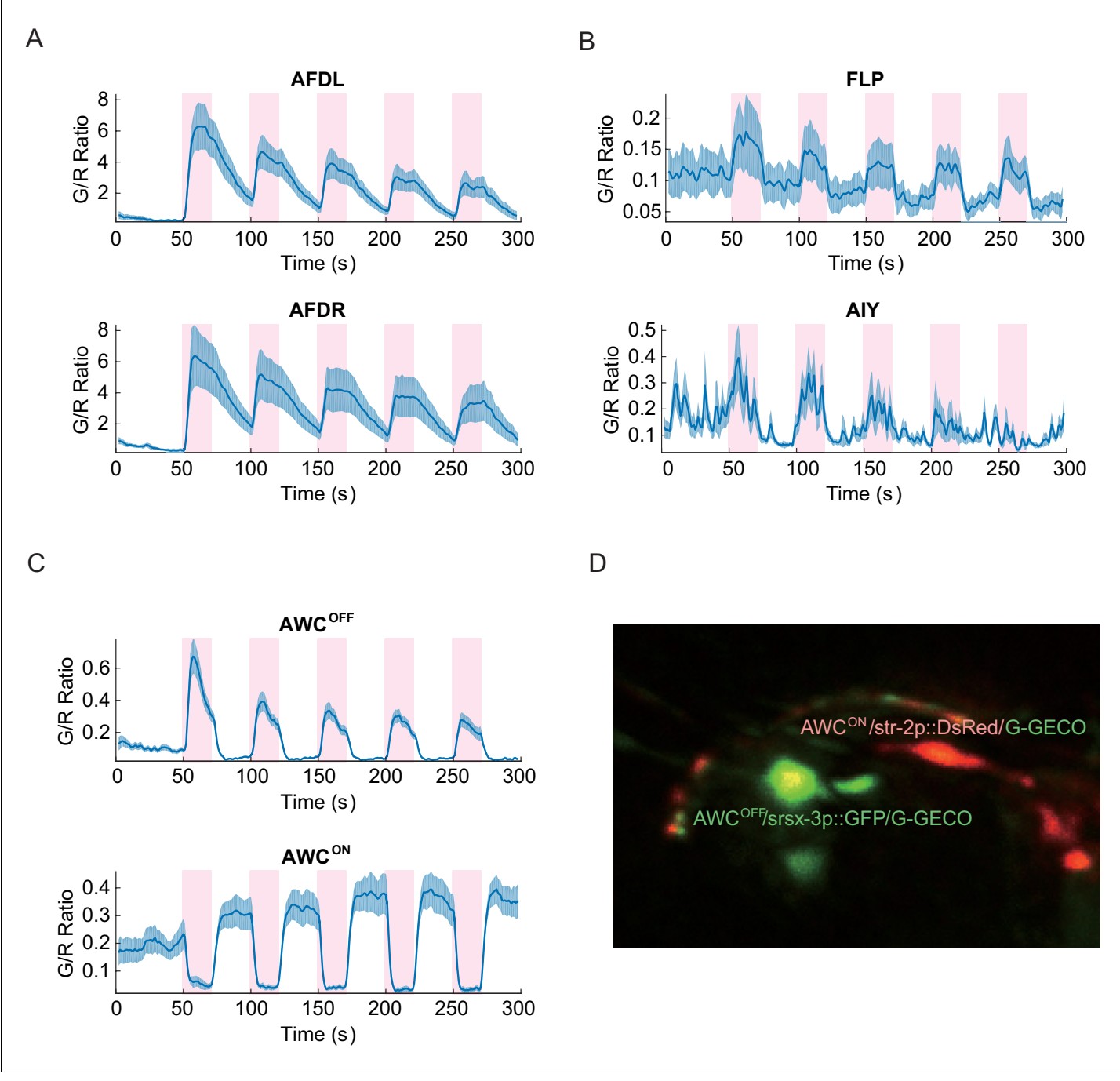

**Figure 2.** Noxious heat stimuli and neural activities in the thermosensory neurons. The pink areas represent the time in which noxious heat stimuli were delivered by a 1440 nm laser. Calcium transients had positive correlation to the laser stimuli in AFDL (**A**, top, $n = 6$, Figure_2-source_data_1.mat), AFDR (**A**, bottom, $n = 7$, Figure_2-source_data_1.mat), FLPs (**B**, top, $n = 10$, Figure_2-source_data_2.mat), AIYs (**B**, bottom, $n = 18$, Figure_2-source_data_3. mat), and AWC[OFF] (**C**, top, $n = 17$, Figure_2-source_data_4.mat). AWC[ON] (**C**, bottom, $n = 19$, Figure_2-source_data_4.mat) showed negative correlation to the stimuli. (**D**) Coexpression of the calcium indicators with AWC[ON]/AWC[OFF] markers for the asymmetry identification. Calcium measurements were made in the nuclei in AFD, AWC, and FLP, and in the neurites in AIY. Error bars are standard errors (shaded areas in light blue).

The following figure supplement is available for figure 2:

**Figure supplement 1.** Cellular identity of AWC[ON] and AWC[OFF] were confirmed by coexpressing cell-fate markers (*str-2p::DsRed* and *srsx-3p::GFP*) along with the calcium indicator.

signals in response to the noxious thermal stimuli. One of the AWC neurons produced a deterministic, positive signal, and showed adaptation to repeated heating in a similar fashion to what we measured in AFD. While the other AWC neuron produced a deterministic negative signal that did not adapt. AWC neurons previously have been implicated in chemosensation (*Bargmann et al., 1993*) and thermosensation (*Kuhara et al., 2008*; *Biron et al., 2008*), but not in noxious thermosensation.

Because the left/right neuronal dependence of the asymmetric calcium signals in AWCs switched randomly from worm to worm, we hypothesized that the asymmetry originated from the functional differences between AWC$^{ON}$ and AWC$^{OFF}$ neurons, which also shows random left/right positioning between individual worms (*Wes and Bargmann, 2001*). To test this hypothesis, we expressed G-GECO calcium indicators in the nuclei of AWC neurons, along with AWC$^{ON}$ and AWC$^{OFF}$ specific fluorescent markers (*str-2p::DsRed* and *srsx-3p::GFP*, respectively) with different emission wavelengths (*Lesch et al., 2009*) (*Figure 2D*). Calcium imaging of the transgenic strain revealed that the AWC$^{OFF}$ signaling was positively correlated with the stimuli (*Figure 2—figure supplement 1*, middle), while AWC$^{ON}$ signaling was negatively correlated (*Figure 2—figure supplement 1*, top). The amplitude of AWC$^{OFF}$ signals adapted (*Figure 2C*, top) with repeated stimuli, similar to the calcium signals in AFD neurons (*Figure 2—figure supplement 1*, bottom), but the signals in AWC$^{ON}$ did not show a pattern of adaptation (*Figure 2C*, bottom).

### *nsy-1* and *nsy-7* mutations alter the functional asymmetry in AWC neurons during noxious thermal stimulation

In order to explore the physiological roles of AWC's functional asymmetry in response to noxious thermal stimulation, we expressed G-GECO calcium indicators in *nsy-1* and *nsy-7* mutants. These strains have cell-fate deficiencies with both AWCs becoming AWC$^{ON}$-like in *nsy-1*, and AWC$^{OFF}$-like in *nsy-7* (*Lesch et al., 2009*; *Troemel et al., 1999*). Calcium imaging in these mutants showed clear disruption of the functional asymmetry in response to the noxious stimulation. Both AWC neurons in *nsy-1* correlated negatively with the stimuli (*Figure 3A*) while both AWCs in *nsy-7* correlated positively (*Figure 3B*). Calcium transients in AFD neurons did not have any abnormality in either of the mutants (*Figure 3—figure supplement 1*).

We then performed behavioral screening in these mutants to determine if we could see behavioral variation due to their differences in AWC neuronal types. To perform these experiments, we built an assay system in which a combination of a halogen lamp and thermoelectric heat pump quickly raised the temperature of the entire assay plate from 23°C to 33°C in about 10 s. During each cycle the noxious heat was maintained for 20 s with an ISI of 30 s, and the cycle was repeated 5 times, roughly matching the temperature stimulus of the calcium imaging above. After the acquisition, the behavior of the worms in each frame was labeled as forward, reverse, omega turn, or pause. For each stimulation phase (pre-heat, heating, and cooling), the fraction of worms performing each behavior was measured (*Figure 4—figure supplement 1*).

The fractions of worms performing the forward (*Figure 4A*, left) and reversal behaviors (*Figure 4A*, middle) during the first heating phase in both mutants did not show any significant difference from that of N2 wild-type; however, the fraction of turning behavior in the *nsy-1* mutant but not for *nsy-7* was significantly reduced (*Figure 4A*, right) suggesting a role of the AWC$^{OFF}$ neuron in avoidance behavior. We also noticed that there is a pattern of adaptation in the fractions of turning and reversal behaviors during the repeated heating phases in N2 wild-type and *nsy-7*, but not in *nsy-1* (*Figure 4B and C*). The result matches the pattern of the calcium transients, in which AWC$^{OFF}$ but not AWC$^{ON}$ showed a similar pattern of adaptation.

We reasoned that, because the behavioral fractions are an averaged result of different types of behavioral transitions, changes in behavior due to having different types of AWC neurons might be hidden in the behavioral fractions. To reveal such subtle differences in behavior, we calculated the mean transition rates between all four behavioral states for all strains. We found that for the *nsy-1* mutant, the forward-to-reversal (FR) transitions during the first heating phase were reduced compared to the N2 wild-type and *nsy-7* (*Figure 4D*). It seems that the *nsy-1* mutation also affects the reversal-to-forward (RF) transition rates (*Figure 4—figure supplement 2*), thus resulting in apparently wild-type reversal fraction. To visualize such complex relations between the behaviors and mutations, we generated network graphs of the major behaviors, in which the nodes represent the behavioral fractions and the edges represent transition rates (*Figure 4—figure supplement 3*). The

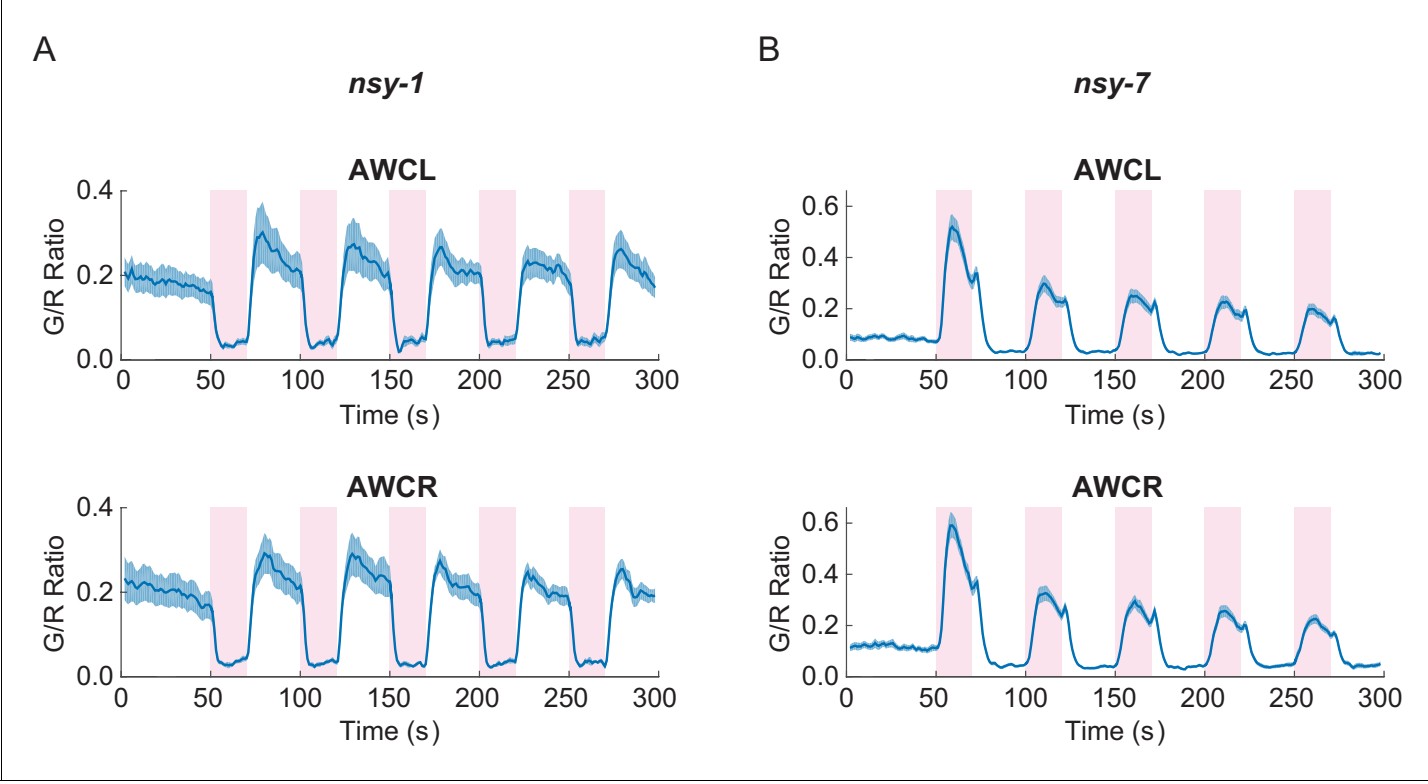

**Figure 3.** Asymmetric neuronal activities by the noxious heat stimulation disappeared in *nsy-1* and *nsy-7* mutants. (**A**, Figure_3-source_data_1.mat) Left and right AWCs became functionally indistinguishable: both AWCL (top, *n* = 10) and AWCR (bottom, *n* = 11) showed calcium transients similar to AWC$^{ON}$ in *nsy-1*. (**B**, Figure_3-source_data_2.mat) In *nsy-7*, AWCL (top, *n* = 20) and AWCR (**B**, bottom, *n* = 21) behaved similar to AWC$^{OFF}$. Pink areas represent laser stimuli. Error bars indicate standard errors (shaded areas in light blue).

The following figure supplements are available for figure 3:

**Figure supplement 1.** Both the *nsy-1* (*n* = 6, Figure_3-source_data_3.mat) and *nsy-7* (*n* = 17, Figure_3-source_data_4.mat) mutations do not affect the activity of AFD neurons in response to the noxious heat stimuli.

**Figure supplement 2.** A 45 min calcium recording in AFD neurons using our pan-neuronal imaging system.

graphs indicate that the apparent reduction of the turn fraction in *nsy-1* is due to the balanced reduction in both the FR and the reversal-to-turn (RT) transitions. Likewise, the apparent similarity in the reversal fraction in *nsy-1* is the result of the reduction in both the FR and the RF transition as mentioned above. The *nsy-1* mutation did not significantly affect the turn-to-forward (TF) transition.

Another interesting feature of the transitions is that the FR transition rates in N2 and *nsy-7* adapt over the repeated heating phases, but not for the *nsy-1* mutant. This pattern is present only in the FR transitions (*Figure 4—figure supplement 2*) and is similar to the adaptation pattern of the calcium transients in AWC neurons where AWC$^{OFF}$ neurons displayed significant adaptation but not the AWC$^{ON}$-like neurons. These results imply that the AWC$^{OFF}$ neuron is responsible for the accelerated FR transition rate during the noxious heating, and for its adaptation to repeated heating stimuli.

## AWC$^{OFF}$ but not AWC$^{ON}$ is required for the normal reversal after noxious thermal stimulation

The plate-heating assays revealed significant reduction in the FR transition rate among the strains lacking functional AWC$^{OFF}$ neurons. In order to reinforce this hypothesis, we focused on this FR transition and sought to observe altered reversal behaviors by inflicting instantaneous noxious stimuli on the worms. In order to perform this task, we have developed a worm-tracking system with a

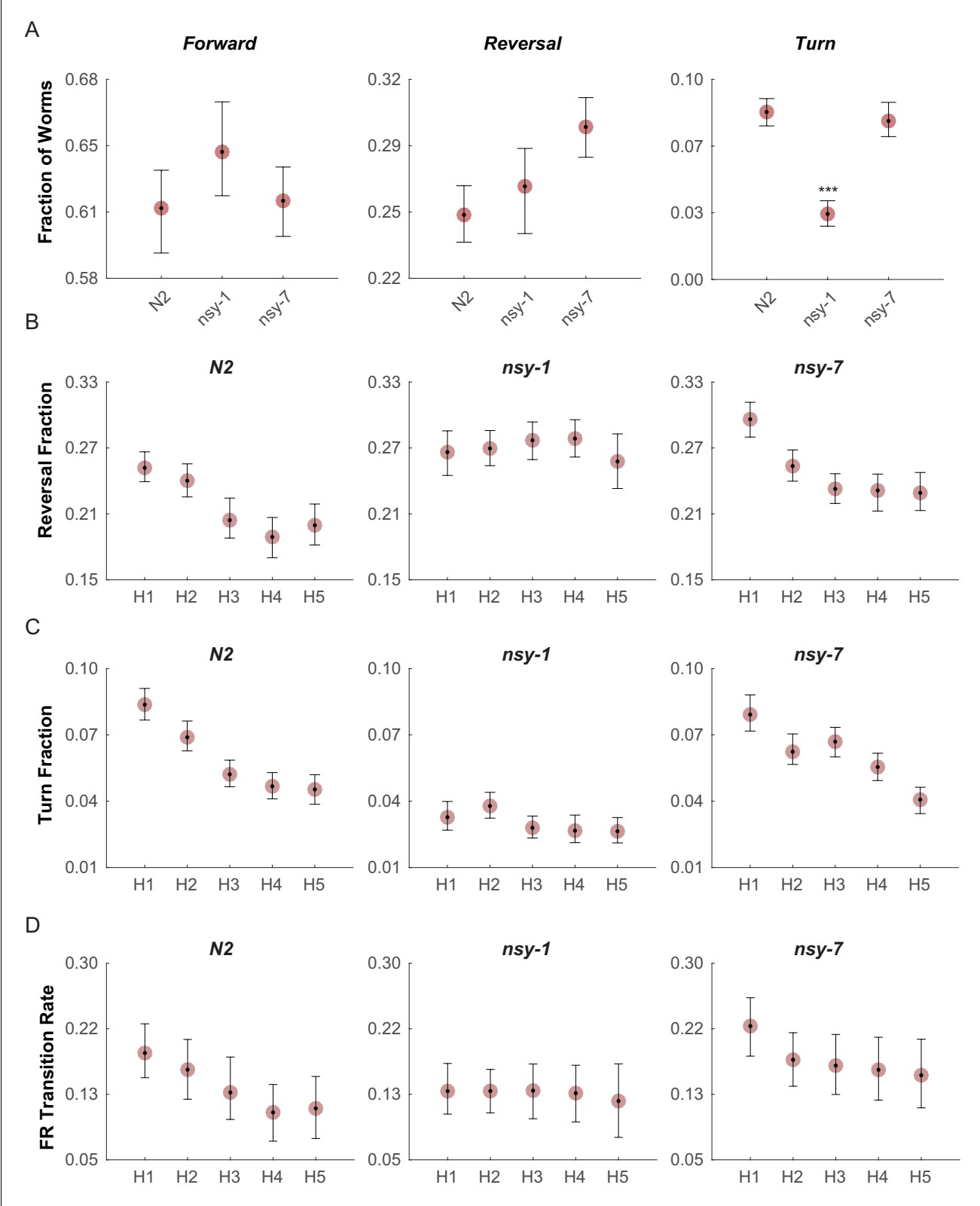

**Figure 4.** Multiple worms were placed on an assay plate, and the temperature of the plate was quickly changed from 23°C to 33°C and back to 23°C for 5 cycles. After the acquisition, the behavior of each worm was carefully labeled as forward, reversal, omega turn, or pause. (**A**, Figure_4-source_data_1. mat) Fraction of worms engaged in each behavior during the first noxious heating phase. During this phase no difference is found in the forward behavioral fraction. A minor difference in the reversal fraction is found between *nsy-7* and the N2. The fraction of turning behavior in *nsy-1* is

*Figure 4 continued on next page*

*Figure 4 continued*

significantly smaller from that of other strains. *** indicates p ≤ 0.001 relative to N2. *n* = 131, 134, and 153 for N2, *nsy-1*, and *nsy-7*, respectively. (**B**, Figure_4-source_data_2.mat) Habituation of reversal and turning behaviors. In both behaviors N2 and nsy-7 but not *nsy-1* display a pattern of habituation during the heating phases. (**C**, Figure_4-source_data_2.mat) Forward-to-reversal transition rates also exhibit habituation in N2 and *nsy-7* but not in *nsy-1*. H1, H2, . . . H5 indicate the heating phases 1 to 5, during which the temperature of the assay plate was raised to 33°C from 23°C (**D**, Figure_4-source_data_3.mat). Error bars indicate 83.4% confidence intervals (non-overlapping error bars of 83.4% CIs indicate p is <0.05) calculated by bootstrapping with bias correlated percentile method and 1000 resampling iterations. Detailed statistics for each data point is provided in the supplementary mat files specified above.

The following figure supplements are available for figure 4:

**Figure supplement 1.** Fraction of all the behaviors in each stimulation phase for all the strains.

**Figure supplement 2.** Transition rate of selected transitions in each stimulation phase for all the strains.

**Figure supplement 3.** Graph representation of worm's major behaviors and their transition rates during the first heating phase.

1440 nm laser and galvanometer scanners on which a freely crawling worm can be programmatically zapped at a precise time and body location. A 100 ms laser pulse (50 mA) heats a small region (150 μm) ~1.2°C above the ambient temperature (23°C). A heating pulse with these parameters to the head region of a worm causes a deterministic noxious avoidance response: a long reversal locomotion usually followed by an omega turn. We tested whether there is any abnormality in this avoidance behavior among the AWC asymmetry mutants.

Compared to the wild-type, the *nsy-1* mutant that does not have functional AWC$^{OFF}$ neurons displayed a 5-fold reduction in mean reversal duration, whereas the *nsy-7* mutant that do carry functional AWC$^{OFF}$ neurons maintained a similar reversal duration (*Figure 5A*). Interestingly, the reduction in the reversal duration in *nsy-1* was reverted by increasing the laser current to 150 mA suggesting that the avoidance behavior is triggered either by a mechanism involving a nociceptive threshold, which is higher in the AWC$^{ON}$-like neurons, or by other signaling pathways.

Lastly, we laser-ablated either one or both of the AWC$^{ON}$/AWC$^{OFF}$ neurons and examined their behavior in response to noxious thermal laser heating, to see if the above outcomes in the mutants are the result of the loss of AWC asymmetry or some other mechanism outside AWC. As expected, we observed that ablating the AWC$^{ON}$ neuron did not significantly alter the reversal duration after the stimulation at 50 mA. However, ablating either AWC$^{OFF}$ or both AWCs significantly reduced the reversal duration immediately after the stimulation (*Figure 5B*), confirming that the AWC$^{OFF}$ neuron is required and sufficient for noxious thermal avoidance following the stimulation with the conditions described above. At higher laser currents, worms with the AWC$^{OFF}$ neuron or both AWC$^{ON}$/AWC$^{OFF}$ neurons ablated increased their reversal duration, suggesting the likely presence of another thermal nociceptive signaling pathway.

## Discussion

In order to understand how a network of neurons function as a system, it is important to be able to characterize the dependence of a behavioral output on the internal states of the network (*Roberts et al., 2016*). Up until now, much effort has been spent on finding such causative correlations by first screening for abnormal behaviors among mutants, making educated guesses as to which neurons were responsible for the abnormality, and then generating transgenic animals that express functional probes such as a fluorescent calcium indicator in the candidate neurons. Only after this lengthy process could one start looking at specific neurons for the evidence of neural mechanisms. If there were no difference in the neural activities of the candidate neurons in the mutants, then the whole process would have to be repeated. Due to the iterative and serial nature of this approach it is an innately inefficient method to map out functional connections.

Our pan-neuronal approach, among other similar approaches previously reported, is more direct than most neurogenetic schemes. The functional screen broadly searches for activity in response to specific perturbations such as externally applied heat stimulation. If correlations between the applied perturbation and neural activities can be identified, it is relatively easy to functionally manipulate the

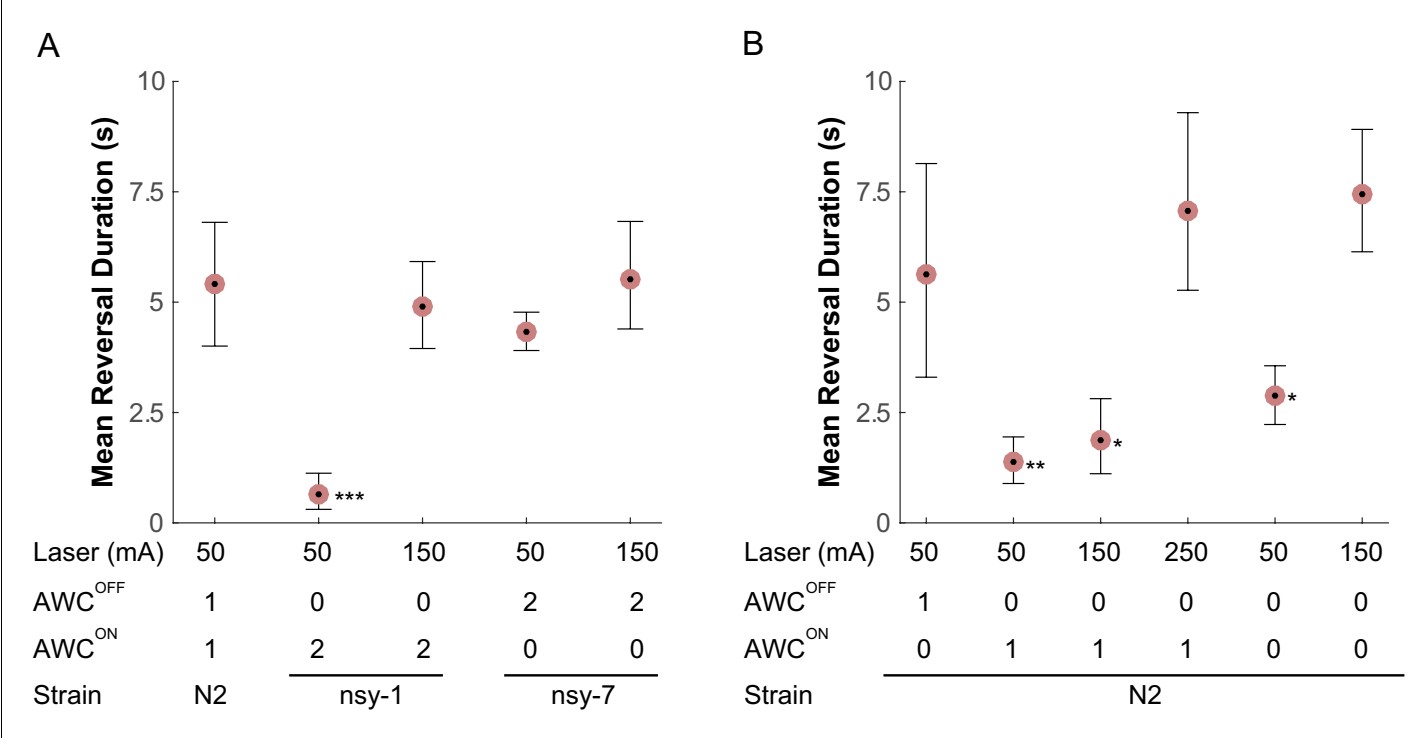

**Figure 5.** AWC asymmetry mutant strains (A) as well as AWC-ablated worms (B) were tested for the avoidance behavior right after a short pulse of noxious stimulation. The acquired images were manually examined to determine the duration of the reversal behavior after a laser zap. The mean reversal duration was plotted for each combination of conditions. Applied laser power, number of functional AWCs either as a result of mutation or laser ablation, and strain names are displayed at the bottom of each plot. Error bars are the 83.4% confidence intervals calculated by bootstrapping. Mann-Whitney was used to compare two means. P-values in relative to the control (A, leftmost) are indicated as follows: *$p \leq 0.05$; **$p \leq 0.01$; ***$p \leq 0.001$. Number of samples from left ($n$ = 12, 16, 5, 11, 14, 5, 19 11, 20, 11, 11). Detailed statistics for each data point is provided in the supplementary mat file (Figure_5-source_data_1.mat).

network by using readily available techniques such as cell-specific mutations, cell ablations, and optogenetics, in order to reveal the neural mechanisms. With a pan-neuronal approach, the critical steps can be performed within the same level of the system's hierarchy, thus facilitating the interpretation of the results. Once the responsible neurons are identified for the perturbation, one can proceed to behavioral assays using similar stimuli, and identify the behavioral outputs to complete the mapping of the entire signal transduction from the external perturbation to behavioral output via the identified neuronal circuitry.

As we have demonstrated in this study, the pan-neuronal approach can be very efficient in that we were able to map most of the previously reported thermosensation circuitry in a single screening. Moreover, we also identified previously unidentified neurons such as RIS for thermosensation and AWC asymmetry for thermal nociception. From the results of the initial functional screening, we chose the AWC asymmetry to further investigate the neural mechanisms of the thermal nociception circuitry.

The behavioral assay results of the *nsy-1* mutant and AWC$^{OFF}$ ablated worms, suggest that the AWC$^{OFF}$ neuron plays a critical role in sensing and/or processing thermal nociception in the central nervous system of *C. elegans*. For the transition rates of the behaviors in the plate-heating experiment, the genetic conversion of AWC$^{OFF}$ to AWC$^{ON}$ resulted in reduced FR, RF, and RT transitions and the mutation also caused an overall reduction of omega turns. Moreover, for the head-directed laser stimulation assay, the duration of the escape response (reversal) was reduced only when AWC$^{OFF}$ was absent. In the previous studies concerning olfactory functions in AWC neurons, AWC$^{OFF}$ was implicated in detecting the removal of attractive odor, and in activating a local search behavior consisting of reversal and omega turns (*Chalasani et al., 2007*). Our findings are in good agreement with the

previous ones in that both in the olfactory and nociceptive sensation, AWC$^{OFF}$ is required for reversal locomotion and omega turns.

The role of the AWC$^{ON}$ neuron in nociception is less clear. Because the *nsy-1* mutant has two functional AWC$^{ON}$ neurons, it was thought that some of the *nsy-1*-specific behavior might have been caused by an increase in total AWC$^{ON}$ activity. This hypothesis was rejected by ablating only the AWC$^{OFF}$ neuron, leaving a single AWC$^{ON}$ neuron, and showing that the reversal duration was essentially the same in both cases. The plate-heating behavioral measurement of *nsy-7* implied a slightly elevated reversal fraction with decreased RF and TF transition rates during nociceptive stimulation. AWC$^{ON}$ might have a role in exiting the avoidance behavior, similar to dispersal behavior in foraging; further study is required for AWC$^{ON}$'s involvement in this role.

There have been some discrepancies reported for the roles of AWC neurons in thermosensation (*Kuhara et al., 2008*; *Biron et al., 2008*). One possible explanation is that since it was not a common practice to distinguish between AWC$^{ON}$ and AWC$^{OFF}$ functions in thermosensation, the calcium signals in these neurons might not be identical. This is only a speculation since we did not detect strong nuclear calcium signals in AWC neurons in the physiological temperatures using our pan-neuronal calcium indicators. Future studies might be able to answer some of the questions by using a temperature range much wider than a typical thermosensation analysis, so that the true dynamic range of AWC's thermosensation, even in the extremes, can be analyzed.

There are some limitations with our approach and a number of improvements to be made. Our imaging is done while the worm is paralyzed and in principle the signals we see here might be different than those in a freely behaving worm. However, this is why we follow up with detailed behavioral measurement using similar thermal stimuli on freely moving worms, so we can connect neuronal function with behavioral output. Recent work has demonstrated measuring calcium signals in freely behaving worms (*Nguyen et al., 2016*; *Zheng et al., 2012*; *Venkatachalam et al., 2016*) and so a completely integrated experiment with the detailed signal and behavioral measurements shown here is certainly possible. Also, our pan-neuronal calcium indicators are localized only to the nuclei and so we miss a number of functional signals that are located in the cytoplasm, perhaps far distal to the cell body. Furthermore, vigorous research is currently taking place to improve the performance of functional fluorescent indicators. Newer indicators such as GCaMP6 might reveal neuronal signals with more detail due to the difference in their physical properties such as calcium dissociation constant, which might be more suitable for some neurons in *C. elegans*. We proceeded here with this limitation because we knew that cellular identification and segmentation would be challenging, and after identifying neuronal candidates we follow up with traditional cytoplasmic calcium indicator expression and measurement. As the computational process of segmentation advances, future research may try to express the calcium indicators in some of the interneurons in other subcellular compartments such as the cell body and neurites. In such scenario, one should perhaps use calcium indicators with different fluorescence wavelengths to differentiate the neural fibers that are located closer than the diffraction limit of visible light. Alternatively, superresolution microscopy may be utilized to resolve the congested neural fibers, so that the pan-neuronal imaging might be practicable without relying on the nuclear expression of the indicators.

In conclusion, we have developed a pan-neuronal imaging-based functional screening scheme, which can concurrently measure the neural activities of most of the neurons in *C. elegans*. We have demonstrated that the screening scheme can efficiently identify the neural circuitry for various sensory mechanisms, both previously known and unknown. Among the newly discovered circuitry, we investigated in detail the roles of AWC asymmetry in thermal nociception, and showed that AWC$^{OFF}$ neuron is critical to initiate the avoidance behavior by accelerating the FR transition rate. This screening scheme increases the efficiency of functionally mapping circuits from sensation to behavioral output. It is our hope that this technology will assist researchers interested in understanding complex neural functions in *C. elegans* and other optogenetically accessible model systems.

## Materials and methods

### Pan-neuronal imaging system with thermal stimulation

An inverted microscope (Eclipse Ti, Nikon, Melville, NY) with two stacked fluorescence filter turrets was configured to acquire two channels of images simultaneously with two EMCCD cameras

(iXon$^{EM}$+ DU-897, Andor, Belfast, UK): the upper turret was used for fluorescence illumination with a regular filter cube; while the lower turret was used for spectrally separating fluorescent images using a special rigid filter cube (91032, Chroma, Bellows Falls, Vermont) with a dichroic mirror (T505lpxr, Chroma). Long-pass images go to a camera port on the left side while the reflected images go to a customized port in the back. An objective lens (Plan Apo 60x WI NA 1.20, Nikon) was equipped with a piezo flexure objective scanner (P-721.SL2, PI, Karlsruhe, Germany) and a digital piezo controller (E-709.SRG, PI) for fast scanning along the z-axis. A fast scanning stage (MLS203, Thorlabs, Newton, New Jersey) with a brushless servo controller (BBD203, Thorlabs) was used for tracking the small positional changes during thermal stimulus. A fiber-coupled IR laser (FOL1404QQM-617-1440, Fitel, Tokyo, Japan) and its controller (LDC210C, Thorlabs) were used for local heating. We chose the wavelength to selectively heat up water molecules without directly stimulating any other biomolecules in the tissue (*Smith et al., 2009*). High-power LEDs (M617L2, M470L2, Thorlabs) along with their drivers (LEDD1B, Thorlabs) were used for fluorescence excitation. A thermoelectric heat pump (MCTE1-19913L-S, Farnell, Leeds, UK), a thermoelectric recirculating chiller (T255P-3CR, Coherent, Santa Clara, California), and a temperature controller (5R6-900, Oven Industries, Camp Hill, Pennsylvania) controlled the temperature of the specimen. An analog output module (NI 9263, National Instruments, Austin, Texas) sent control signals to the LEDs, laser controller, and piezo controller. An analog input module (NI 9219, National Instruments) was used for feedback acquisition from the piezo controller and laser controller for their positions and power, respectively. All the controllers, drivers, cameras, and the microscope were connected to a PC (Windows 7 64-bit, Microsoft, Redmond, Washington); a custom-made software package in MATLAB (Mathworks, Natick, Massachusetts) was used for coordinated illumination, z-axis scanning, multi-channel image acquisition, sample movements, temperature control, noxious thermal stimulation, auto-focusing, and data recording. The Nikon Ti SDK (4.4.1.728 64bit) and the ImageJ API (*Schneider et al., 2012*) were used to bridge the microscope and cameras to the MATLAB scripts. The software package is available here: https://github.com/ikotera/WormAnalyzer.

## Calcium imaging

A transgenic adult worm was anesthetized with 20 mM levamisole (31742, Sigma-Aldrich, St. Louis, Missouri) and sandwiched between a coverslip and a 300 µm-thick 2% agarose pad on a microscope slide. We waited 15 min before recording to let the worm's muscle tension come to equilibrium. The temperature of the sample was calibrated and maintained with a custom-designed temperature control system, which monitored the temperature of the room, objective lens, and microscope slide, and adjusted the thermoelectric elements accordingly. To minimize phototoxicity and maximize image quality and time resolution, the LED and camera shutter were synchronized so that the worm was illuminated only during the CCD exposure (10 ms, 10–20 fps). For calcium imaging with noxious heat stimulation, the images are acquired for 50 s without any stimulus; then the predefined heat stimulus for 20 s with an ISI of 30 s and repeat count of 5 were performed. Most of the worms survived this condition and showed no behavioral defect a few hours after they were transferred back to an NGM plate.

## Image analysis

A software package was written in MATLAB for all of the image processing (all scripts except for the 3D deconvolution code available here: https://github.com/ikotera/WormAnalyzer). The image streams from two cameras were saved on a disk as raw binaries. The analysis script loads all the data to RAM (minimum 16GB required), and performs 3D deconvolution on a CUDA-capable GPU (GTX770, NVidia, Santa Clara, California) by a custom-made algorithm (*Bruce and Butte, 2013*). Then the images are subjected to subpixel image registration by a single-step discrete Fourier transform algorithm (*Guizar-Sicairos et al., 2008*) utilizing the GPU. After the pretreatment of the images, the script adaptively segments and tracks all the neurons, and extracts calcium signals from each neuron. The analyzed images and data are saved as binaries. The process up to here is fully automatic without any human intervention, and takes about 3x the image acquisition time with a high-end CPU and GPU. We designed a GUI to help with manual identification. The GUI is loaded for displaying images and neural activities with an easy navigation through multiple image planes

and neurons by keyboard shortcuts and mouse clicks, vastly improving the efficiency of manual neuronal identification.

## Neuronal segmentation, measurement, and miscellaneous calculations

After the image analysis process, neural signals were quantized automatically by a gradient vector field (GVF)-based cell-tracking algorithm (*Li et al., 2007*). Briefly, starting from all the pixels in the first image, the computed GVF is used to find the initial location of all the neurons. A small region is cropped out from the center of each neuron in the next image, and the local flow tracking is performed in the cropped region. The local flow tracking is repeated until the last image for all the initial neurons. The acquired time series were resampled by linear interpolation for statistical analyses.

**Table 1.** A list of worm strains used in this study.

| Serial Number | Strain Name | RRID | Genotype | Note |
|---|---|---|---|---|
| 120 | CX10231 | | *kyIs408[srsx-3::GFP;str-2::dsRed2;elt-2::GFP]; nsy-7 (tm3080)* | A gift from Bargmann lab |
| 359 | WSR85 | RRID:WB_WSR85 | *kyIs408; nsy-7(tm3080); rgaIs1; rgaIs2; rgaIs3* | Genotyped for *tm3080*, screened for transgenics |
| 360 | WSR86 | RRID:WB_WSR86 | *kyIs408; nsy-7(tm3080); rgaIs1; rgaIs2; rgaIs3* | Genotyped for *tm3080*, screened for transgenics |
| 363 | VC390 | RRID:WB_VC390 | *nsy-1(ok593)* | CGC |
| 365 | AU3 | RRID:WB_AU3 | *nsy-1(ag3) II* | CGC |
| 366 | CX7894 | RRID:WB_CX7894 | *kyIs408* | A gift from Bargmann lab |
| 370 | WSR90 | RRID:WB_WSR90 | *kyIs408; rgaIs1[rgef-1p::NLS-G-GECO1.1-T2A-NLS-DsRed2]; rgaIs2[rgef-1p::NLS-G-GECO1.1-T2A-NLS-DsRed2]; rgaIs3[tax-4p::NLS-mNeptune]* | Generated by crossing CX7894 and WSR90 |
| 372 | WSR92 | RRID:WB_WSR92 | *rgaIs1[rgef-1p::NLS-G-GECO1.1-T2A-NLS-DsRed2]; rgaIs2[rgef-1p::NLS-G-GECO1.1-T2A-NLS-DsRed2]; rgaIs3[tax-4p::NLS-mNeptune]* | Screened for transgenics |
| 373 | WSR93 | RRID:WB_WSR93 | *rgaIs1[rgef-1p::NLS-G-GECO1.1-T2A-NLS-DsRed2]; rgaIs2[rgef-1p::NLS-G-GECO1.1-T2A-NLS-DsRed2]; rgaIs4[glr-1p::NLS-mNeptune]* | Screened for transgenics |
| 384 | WSR99 | RRID:WB_WSR99 | *kyIs140 [str-2::GFP + lin-15(+)]; nsy-1(ky397); rgaIs1; rgaIs2; rgaIs3* | Genotyped for *ky397*, screened for transgenics |
| 385 | WSR100 | RRID:WB_WSR100 | *nsy-1(ok593); rgaIs1; rgaIs2; rgaIs3* | Genotyped for *ok593*, screened for transgenics |
| 406 | WSR120 | RRID:WB_WSR120 | *rgaEx1[ttx-3p::G-GECO1.1-T2A-DsRed2]; rgaEx2[odr-1p::NLS-G-GECO1.1-T2A-NLS-DsRed2; lin-44p::DsRedT3]* | Extra-chromosomal |
| 410 | WSR124 | RRID:WB_WSR124 | *rgaIs5[ttx-3p::G-GECO1.1-T2A-DsRed2; odr1-p::NLS-G-GECO1.1-T2A-NLSDsRed2; lin-44p::DsRedT3]* | 5033 cGy irradiation of strain WSR120 |
| 411 | WSR125 | RRID:WB_WSR125 | *rgaIs6[ttx-3p::G-GECO1.1-T2A-DsRed2; odr1-p::NLS-G-GECO1.1-T2A-NLSDsRed2; lin-44p::DsRedT3]* | 5033 cGy irradiation of strain WSR120 |
| 413 | WSR126 | RRID:WB_WSR126 | *rgaIs5* | WSR124 was outcrossed 2X with N2, resulting in strain WSR126 |
| 414 | WSR127 | RRID:WB_WSR127 | *kyIs408; rgaIs5* | Screened for transgenics |
| 415 | WSR128 | RRID:WB_WSR128 | *kyIs408; rgaIs5* | Screened for transgenics |
| 416 | WSR129 | RRID:WB_WSR129 | *nsy-7 (tm3080); rgaIs5* | Genotyped for *tm3080*, screened for transgenics |
| 418 | WSR131 | RRID:WB_WSR131 | *kyIs408/+; rgaIs5/+* | Screened for transgenics |
| 420 | WSR133 | RRID:WB_WSR133 | *nsy-1(ok593); rgaIs5* | Genotyped for *ok593*, screened for transgenics |
| 421 | WSR134 | RRID:WB_WSR134 | *nsy-1(ok593); rgaIs5* | Genotyped for *ok593*, screened for transgenics |

**Table 2.** A list of plasmid constructs used in this study.

| Plasmid Name | Content | Plasmid Construction | Donor Vecotr #1/ att-PCRP-att | Donor Vector #2 | Donor Vector #3 | Donor Vector #4 |
|---|---|---|---|---|---|---|
| pWRPN01 | {pENTR L1-odr-1 promoter-L5r} | BP reaction: att-odr-1 promoter-att + pDONR P1-P5r | attB1-odr-1 promoter-attB5r | pDONR P1-P5r | | |
| pWRPN02 | {pDEST R1-chloramphenicol-ccdB-R2/pPD95.75} | pPD95.75 was cut with AgeI and EcoRI to remove GFP, then Gateway cassette RfA was inserted; | | | | |
| pWRPN03 | {pENTR L1-ttx-3 element-L4} | BP reaction: att-ttx-3 element-att + pDONR P1-P4 | attB1-ttx- 3 element-attB4 | pDONR P1-P4 | | |
| pWRPN04 | {pENTR L5-DsRedT3-L2} | BP reaction | | | | |
| pWRPN05 | {pENTR L1-lin-44 promoter-L5} | BP reaction | | | | |
| pWRPN06 | {pENTR L5-NLS-G-GECO1.1-T2A-NLS-DsRed2-L2} | BP reaction: att-NLS-G-GECO1.1-T2A-NLS-DsRed2-att + pDONR P5-P2 | attB5-NLS-G-GECO1.1-T2A-NLS-DsRed2-attB2 | pDONR P5-P2 | | |
| pWRPN07 | {pExp B1-lin-44 promoter-B5-DsRed.T3-B2/pP95.75} | LR reaction: pWRPN05 + pWRPN04 + pWRPN02 | pWRPN05 | pWRPN04 | pWRPN02 | |
| pWRPN08 | {pExp odr-1 promoter-NLS-G-GECO1.1-T2A-NLS-DsRed2/pP95.75} | LR reaction: pWRPN01 + pWRPN06 + pWRPN02 | pWRPN01 | pWRPN06 | pWRPN02 | |
| pWRPN09 | {pEXP ttx-3 element-G-GECO1.1-T2A-DsRed2/pPD95.75} | LR reaction: pWRPN03 + pENTR L4-G-GECO1.1-L3 + pENTR L3-DsRed2-L2 + pWRPN02 | pWRPN03 | pENTR L4-G-GECO-L3 | pENTR L3-DsRed2-L2 | pWRPN02 |
| pWRPN10 | {L3613 rgef-1 promoter-NLS-G-GECO1.1-T2A-NLS-DsRed2} | | | | | |
| pWRPN11 | {L3613 tax-4 promoter-mNeptune} | | | | | |
| pWRPN12 | {L3613 glr-1 promoter-mNeptune} | | | | | |

Mean calcium activity and standard errors were calculated after they were normalized to the minimum of the time series, which is usually the baseline of the transients. For population behavioral analyses, transition time points were extracted first, then the length of behavior immediately before the transition was measured. Inverse of the pre-transition duration is the rate of transition. Average transition rates were calculated by harmonic mean, and corresponding standard deviation by jackknife estimation (*Lam et al., 1985*). For statistical comparison of two means, we employed the nonparametric Mann-Whitney test to see if they have significant difference. As for the error bars in the plots, we used 83.4% confidence interval to visualize the statistical difference of the means (nonoverlapping 83.4% CIs correspond to P-value being <0.05).

## Population behavioral assays

We equipped a stereo microscope (MVX10, Olympus, Tokyo, Japan) with a halogen lamp illuminator (MI-150, inner IR filter removed, Dolan-Jenner Industries, Boxborough, Massachusetts), a thermoelectric heat pump (MCTE1-19913L-S, Farnell), and a temperature controller (5R6-900, Oven Industries). These devices were used synchronously to rapidly heat a 10 cm agar pad for the delivery of noxious heat stimuli. A sCMOS camera (pco.edge, PCO, Kelheim, Germany), an LED illuminator (120LED, X-Cite), and the temperature controller were connected to a PC and controlled by a software package (available upon request) written in MATLAB. Temperature of the agar was carefully calibrated, monitored, and automatically adjusted during the assay. About 20 worms were washed twice with M9 buffer matching the $T_C$, and carefully transferred to the agar pad by a siliconized pipette tip. Excess liquid was removed by gentle puff of nitrogen. We let worms crawl around for 5 min before the start of acquisition. The image stream was saved as a raw binary on an SSD RAID system for maximum possible frame rate and resolution (20 fps, 2048 x 2048). A GUI based script was

**Table 3.** A list of genotyping performed in this study.

| Mutation to Genotype | Forward Primer 5' –> 3' | Reverse Primer 5' –> 3' | Sequencing Primer | PCR Conditions* | Wild-type | Mutant |
|---|---|---|---|---|---|---|
| gcy-8 (oy44) | (WRO245) gcctaccaaatta tttcaaacatc | (WRO246) TTGATAATTAAA ATGCAAGACGAAC | N/A | Phusion Hot Start II; 1. 98°C, 1:00; 2. 98°C, 0:15; 3. 58°C, 0:30; 4. 72°C, 1:45; 5. Go to 2, 34x; 6. 72°C, 10:00; 7. 4°C | 2225 bp band | 750 bp < band < 1 kb |
| gcy-18 (nj38) | (WRO247) GAATAGAATGAGA CGAATGAAATTTG | (WRO248) TGTTACCTACCAAGT GCCTAACTTAC | N/A | Phusion Hot Start II; 1. 98°C, 1:00; 2. 98°C, 0:15; 3. 58°C, 0:30; 4. 72°C, 1:45; 5. Go to 2, 34x; 6. 72°C, 10:00; 7. 4°C | 1459 bp band | ~500 bp band |
| gcy-23 (nj37) | (WRO252) CATCTACGGC TACATCCATCTC | (WRO253) TCCATCATACGC ATCATCTG | N/A | Phusion Hot Start II; 1. 98°C, 1:00; 2. 98°C, 0:15; 3. 65°C, 0:30; 4. 72°C, 1:45; 5. Go to 2, 34x; 6. 72°C, 10:00; 7. 4°C | 2141 bp band | 1 kb < band < 1.5 kb |
| nsy-1 (ky397) (WRO243) agtcagccatcaa gtcctattg | | (WRO244) TTTCAACCAACC TGGCC | (WRO268) CGATGATAC AAATCACC | Phusion Hot Start II; 1. 98°C, 1:00; 2. 98°C, 0:15; 3. 56.9°C, 0:30; 4. 72°C, 0:20; 5. Go to 2, 34x; 6. 72°C, 10:00; 7. 4°C | atagttcaagtcgattcttcatg cttCaaaaggattca gaacgtagaagatc | atagttcaagtcgattcttcatg cttTaaaaggattca gaacgtagaagatc |
| nsy-1 (ok593) | (WRO269) agattcatcaatc cgagttg | (WRO270) CGAACTCGTTCT TCACGAC | N/A | Phusion Hot Start II; 1. 98°C, 1:00; 2. 98°C, 0:15; 3. 58°C, 0:30; 4. 72°C, 1:45; 5. Go to 2, 34x; 6. 72°C, 10:00; 7. 4°C | ~2.5 kb band | ~280 bp band |

*Table 3 continued on next page*

*Table 3 continued*

| Mutation to Genotype | Forward Primer 5' –> 3' | Reverse Primer 5' –> 3' | Sequencing Primer | PCR Conditions* | Wild-type | Mutant |
|---|---|---|---|---|---|---|
| *nsy-7 (tm3080)* | (WRO237) atgggataaggtt ggtaactagc | (WRO238) TACAGGTTGCG AAAGGATATTC | N/A | Phusion Hot Start II; 1. 98°C, 1:00; 2. 98°C, 0:15; 3. 65°C, 0:30; 4. 72°C, 1:30; 5. Go to 2, 34x; 6. 72°C, 10:00; 7. 4°C | ~700 bp band | ~225 bp band |
| *osm-9(ky10)* | (WRO235) GATTATATCA AATGGAAGAAGGGAG | (WRO236) GAGTCCTGGAG ATTCGGG | (WRO255) AACAAGCGG CAAATGCTAGG | Phusion Hot Start II; 1. 98°C, 1:00; 2. 98°C, 0:15; 3. 58.8°C, 0:30; 4. 72°C, 1:30; 5. Go to 2, 34x; 6. 72°C, 10:00; 7. 4°C | ctttaggc**C**aatcagccctcc | ctttaggc**T**aatcagccctcc |
| *ttx-1 (p767)* | (WRO191) ccaaatttcaaaa tttgagcactcaa aactctgcct | (WRO193) GTAGATTCCGAA TTTGCTAGTGGT AACGTCC | (WRO196) TTCTGGGAT TTTTCAGACTTTCC | Phusion Hot Start II; 1. 98°C, 1:00; 2. 98°C, 0:15; 3. 72°C, 0:30; 4. 72°C, 0:40; 5. Go to 2, 34x; 6. 72°C, 10:00; 7. 4°C | atgaacagcggaaattttt**G**tgg gtttttaaaattaa | atgaacagcggaaattttt**A**tgg gtttttaaaattaa |
| *tax-4 (p678)* | (WRO194) CCTACGACGA AAAAATCAG GTGCATACGAC | (WRO195) GGTCCAATGAG ATCGTTGAATAC TTGTCGAGC | (WRO197) TCAGGTGCA TACGACTACG | Phusion Hot Start II; 1. 98°C, 1:00; 2. 98°C, 0:15; 3. 72°C, 0:30; 4. 72°C, 0:40; 5. Go to 2, 34x; 6. 72°C, 10:00; 7. 4°C | gcggccaccggtggt**C**agccg gcatcttccga | gcggccaccggtggt**T**agccg gcatcttccga |

*Using Thermo Scientific Phusion Hot Start II High-Fidelity DNA Polymerase

developed to quickly go through the image stream to manually label the worm's behavior frame by frame.

## Single-worm behavioral assays

On an optical table, a CCD camera (Manta G-125, Allied Vision Technologies, Newburyport, Massachusetts), a motorized microscope stage (MAX20X, Thorlabs), and a stepper motor controller (BSC102, Thorlabs) were assembled to move a 10 cm agar plate while acquiring images. Two galvanometer scanners (GVS002, Thorlabs), an analog output module (NI 9263, National Instruments), a fiber-coupled IR laser (FOL1404QQM-617-1440, Fitel), a laser diode controller (LDC240C, Thorlabs), and a temperature controller (TED350, Thorlabs) were added to steer and deliver an IR laser beam with high precision. A laboratory-developed software package in LabVIEW (National Instruments) controls all the devices, acquires images, recognizes a freely-moving worm, continuously moves the stage for tracking, and steers galvanometer scanners to deliver IR zaps to the head region of the

worm. 100 ms, 50–250 mA laser zaps were used as noxious heat stimuli. After the acquisition, the images were manually labeled to calculate mean reversal durations right after the laser stimuli.

## Worm strains, transgenes, microinjection, and cell ablations

All strains were maintained according to a standard protocol (*Sulston and Hodgkin, 1988*). Most of the transgenes used in this study were prepared either by PCR fusion (*Hobert, 2002*) or the Gateway system (MultiSite Gateway, Invitrogen, Waltham, Massachusetts). For a list of strains, recombination sites, genotyping conditions, and primers used in this study, see *Tables 1–3*. Microinjection was performed essentially same as previously established (*Mello and Fire, 1995*), with a micromanipulator (MO-202U, Narishige, East Meadow, New York), and microinjector (PLI-100, Harvard Apparatus, Holliston, Massachusetts) on an inverted microscope (TE2000E, Nikon). The injection needle (fire polished aluminosilicate glass with filament, O.D.: 1.0 mm and I.D.: 0.64 mm, AF100-64-10, Sutter) was pulled with a micropipette puller (P-97, Sutter, Novato, California) with the following parameters (P = 999, Heat=482, Vel = 50, and Time = 250): this combination produces vastly superior injection needles both in durability and sharpness to conventional borosilicate ones described elsewhere. Cell ablation assays were performed essentially same as previously reported (*Bargmann and Avery, 1995*), with a UV laser (DUO-220, Spectra-Physics, Mountain View, California), laser dye (10 mM coumarin 440, A9891, Sigma), and an objective lens (Plan Apo 100x Oil NA 1.40, Nikon). Transgenic L2-L3 worms were transferred to a coverslip and anesthetized with 20 mM levamisole (31742, Sigma-Aldrich). A small number (100–1000) of 25 μm microbeads (07313, Polysciences, Warrington, Pennsylvania) were added as spacers between the coverslips to prevent the squashing of the worms. We found this method to be better than the traditional technique, which uses a high concentration of agarose, because it had slower dehydration levels and was easier to set up. Because AWC neurons are located in the outermost region, it is easier and effective to kill the neuron closer to the objective first, and then simply flip the coverslip sandwich for the neuron on the other side.

## Acknowledgements

We would like to thank Byron Wilson for his work on the tracking laser zap system. We also would like to thank Dr. Roger Tsien and Dr. Robert Campbell for kindly providing us the mNeptune plasmid, Dr. Manish J Butte for the libraries for GPU-based deconvolution, and Dr. Cori Bargmann for the CX7894 and CX10231 strains. This research is supported by the Natural Sciences and Engineering Research Council of Canada (WSR and JBR) and Human Frontiers Science Program (WSR, NAT, DF and IK).

## Additional information

### Funding

| Funder | Author |
| --- | --- |
| Natural Sciences and Engineering Research Council of Canada | Jarlath Byrne Rodgers<br>William S Ryu |
| Human Frontier Science Program | Ippei Kotera<br>Nhat Anh Tran<br>Donald Fu<br>William S Ryu |

The funders had no role in study design, data collection and interpretation, or the decision to submit the work for publication.

### Author contributions

IK, Conception and design, Acquisition of data, Analysis and interpretation of data, Drafting or revising the article; NAT, DF, Strain construction; JHJK, Contributing the segmentation code, Analysis and interpretation of data; JBR, Performed the behavioral experiments, Acquisition of data; WSR, Conception and design, Analysis and interpretation of data, Drafting or revising the article

**Author ORCIDs**

Jarlath Byrne Rodgers, http://orcid.org/0000-0001-5395-9950

William S Ryu, http://orcid.org/0000-0002-0350-7507

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
