## [Decision Letter]

Thank you for submitting your article "Pan-neuronal screening reveals asymmetric neuronal dynamics of AWC neurons is critical for thermal avoidance behavior" for consideration by *eLife*. Your article has been favorably evaluated by Eve Marder (Senior Editor) and three reviewers, one of whom is a member of our Board of Reviewing Editors. The reviewers have opted to remain anonymous.

The reviewers have discussed the reviews with one another and the Reviewing Editor has drafted this decision to help you prepare a revised submission.

As you can see in the appended reviews below, all three reviewers value the quality and implication of your work. However, there were some substantial concerns about how this work has been framed and a number of other issues require clarification. None of these requested revisions require additional experimentation. Please address each reviewer’s point in detail. In no order of importance the requested changes are:

1) The authors lay claim to a "novel" method for neural-activity screening although the concept is obvious to those in the field and arguably has been "pioneered" – if one can call it that – previously by several other groups. This claim to a novel screening method mars an otherwise great scientific publication. The paper should instead strictly focus on the scientific findings of asymmetric roles for the AWCs in thermal nociception. Specifically, it is recommended that the authors remove the claim of novelty in developing a "pan-neuronal functional screening system" (Introduction, second paragraph) by looking at pan-neuronal calcium responses in response to stimuli. The concept is obvious but, avoiding such discussions, this "technique" was pioneered initially by Kato et al. in their 2015 Cell publication "Global Brain Dynamics Embed the Motor Command Sequence of *Caenorhabditis elegans*" wherein a pan-neuronal calcium signal is screened during fictive locomotory behavior and the active neurons identified through fluorescent reporters (Figure 1 of the Kato paper).

Furthermore, Venkatachalam et al. apply the very same idea to the worm thermosensory circuit in their 2015 publication "Pan-neuronal imaging in roaming *Caenorhabditis elegans*" uncovering 2 unnamed neurons labeled 62 and 18 (Figure supplement 3 of the Venkatachalam paper). A citation which is strikingly missing from the submitted paper despite being published in PNAS within the same issue as the Nguyen paper which Kotera et al. do cite. Further to this, Kotera and colleagues should cite the work of the Samuel's lab as it is a direct precedent to their own work and sets the stage for their newly published discoveries.

Altogether, the claim of a novel technique is a distraction from what is otherwise a great scientific paper.

2) The authors use several fluorophores that present odd choices given the availability of newer and better alternatives. This is not problematic but readers will be left to wonder why such choices were made. We recommend the authors address this upfront to direct readers to the alternatives.

Chief among these strange choices is GECO1.1. While an improvement on GCaMP3, GECO1.1 fairs poorly when compared to GCaMP6 (the new standard in the field) – please see both the original papers from Zhao et al. 2011 on GECO1.1, Chen et al. 2013 on GCaMP6, and the Neurophotonics comparison in 2015. Presumably the choice was influenced by when Kotera and colleagues began their work. Still, given the array of choices, readers should be advised of best practices.

Second, the Discosoma-derived proteins such as DsRed, DsRed2, and mCherry have long been known to have ill effects in worm and, recent yet to be published work by Monica Driscoll's lab show that neurons react poorly to these fluorescent proteins. For this reason, TagRFP is often used in place of DsRed.

Third, mNeptune has been superseded by mNeptune2.5 which provides a nearly 2-fold improvement in brightness – published in 2014 by the Lin lab at Stanford.

As stated, these are not major issues. The authors should simply address choices for best practices since readers have easy access to strains and plasmids that represent newer better alternatives to the ones used.

3) The authors discover several unreported neurons to be thermo-sensitive. They choose to focus on AWC, a neuron with multiple but conflicting publications stating its thermo-sensitivity. The readers are left to wonder why the authors forewent the more obvious choice of exploring the novel finding of unreported thermo-responsive neurons. The authors should address this choice. Perhaps another paper is forthcoming with their results for RIS. RMDV and SMDV activity (motor neurons innervating the head) are likely reflections of downstream head actions in response to thermal stimulus. This a simple loose end that can receive a quick mention.

4) The authors have developed an impressive imaging rig capable of delivering thermal stimulus. Presumably a future paper will cover this novel rig but for now, in the spirit of openness, the software should be released to the community for public use in an open source repository and not simply made available upon request. This also benefits the science by providing transparency in the algorithms used to identify neurons, resolve their activity, and the analysis used to assay neural activity and behavioral correlates.

5) The supplement should match up the reporters used in the paper with the neurons they were used to identify. As has been the case multiple times in our field, future papers may find that some of the identifications were erroneous. This will help in quick corrections to the knowledge base.

6) G/R ratio should be explicitly defined in Figure 1, where it is first used, as opposed to Figure 2.

7) In Figure 1—figure supplement 2, the thermal stimulus should be marked on the individual graphs so as to make sense of the neural traces.

8) Figure 2—figure supplement 1 shows the poor S/N in AWC-OFF due to the G-GECO measurement against the srsx-3p::GFP reporter. The S/N vastly improves in Figure 2. How was this accomplished? Did the authors interpolate AWC OFF and ON from their thermal responses and then measure their activity with no OFF/ON identification reporters present?

9) The AWC fates in nsy-1 and nsy-7 mutant background requires a citation (subsection “nsy-1 and nsy-7 mutations alter the functional asymmetry in AWC neurons during noxious thermal stimulation”, first paragraph).

10) The term "turn" (first mentioned in the second paragraph of the subsection “nsy-1 and nsy-7 mutations alter the functional asymmetry in AWC neurons during noxious thermal stimulation” and Figure 4) can have multiple behavioral definitions for worms. The authors should be explicit as to what they mean by "turn". Are these omega turns, short reversals that include turning behavior, simple left/right turns?

11) The type of statistical tests should be explicitly stated and justified alongside the data – not just in the Methods – so that readers can assess the implicit assumptions made when comparing measured distributions. Furthermore, claims of normal distributions should be backed by histograms or similar representations of the sampling. Worm behavior often deviates from normality and therefore non-parametric tests are often a more appropriate choice. This problem occurs in Figure 4, Figure 4—figure supplement 3, and Figure 5. The authors can switch to violin plots or similar such statistical representations to assure the readers that the sampling is indeed "normal".

12) There is no N provided for Figure 4. The sample size should be explicitly stated. Furthermore, reversal rate and pausing (Figure 4—figure supplement 1) appear to be different in a nsy-7 mutant background. Did the authors repeat the experiments with larger sample sizes to rule out a role for AWC-ON in reversal rate and pausing as a response to noxious thermal stimulus? The behavioral transition graphs in Figure 4—figure supplement 3 show a clear non-wildtype role for nsy-7 and, inferentially AWC-ON, in response to noxious heat. Yet, these findings receive only a minor mention in the Discussion. We would like some mention at the location where the data is shown as well.

13) Claims of behavioral adaptation (and lack thereof) to repeated thermal stimulus in Figure 4 should be backed by a goodness of fit regression to a linear or exponential model of adaptation – or a similar statistic. Scientific claims require stronger evidence than that presented.

14) On the figures, the error bars are labeled as 83.4% confidence level but presumably the CI is 95% and 83.4% is termed the corresponding "capture percentage". The correct term should be used.

15) Figure 5 fails to show not only the N for the variety of conditions tested but, also, the WT response to 150mA and 250mA laser stimulation. This leads to questions as to how AWC ON/OFF ablation was controlled in the statistical analysis of 150mA and 250mA laser stimulation. The authors should address this by showing the missing data and explaining how the statistical tests were performed.

16) The term "sedated" (Discussion, seventh paragraph) is inappropriate for levamisole-mediated paralysis. The worm's neurons are obviously still functional after application of levamisole. We suggest using the term "paralyzed" in place of "sedated".

17) The central claim that AWC_OFF was identified by whole brain recording is powerful, but the only data that shows whole brain recording in Figure 1 is in a regime where AWC_OFF is quiet. It would be helpful to show the whole brain recording data that actually uncovered AWC.

18) An impressive number of cells was identified. They mention using glr-1 expression patterns to help with cell identification, but a more detailed explanation of how they came to their cell identities would be useful.

19) In the text, they claim to have stable recordings for 60 minutes with this technique. This is an impressive claim, and should be supported by data.

20) Figure 5 has a typo. The data that correspond to the ablation experiments should be labeled as such.

21) The manuscript was overall quite poorly written which made it difficult to read. The rationalization about different 'hierarchies' in investigating mechanistic underpinnings of behavior was very unclear. After all, calcium imaging is hardly the best or most direct readout of neuronal activity. This was an issue in the Introduction but even more so in the Discussion. This is not the first paper in *C. elegans* or in any other system to infer neuronal functions from examining stimulus-evoked neuronal activity, and it is inaccurate to portray it as such and to not mention many previous similar reports (for *C. elegans* – for instance see work from the de Bono lab, Chalasani lab etc.).

22) The authors also do themselves a disservice by not discussing the previous pan-neuronal imaging papers in more detail. Papers from other labs reporting similar imaging methods should be introduced in more detail. In particular, the paper by Venkatachalam et al. from the Samuel lab which specifically reports pan-neuronal imaging of thermal stimuli in freely moving animals is not referenced at all.

23) Related to the above, please check references throughout. In many cases, references are missing altogether or the wrong references are included. For example, the role of AFD in thermosensation (Introduction, fourth paragraph) was first shown by Mori et al. in 1995. The Biron et al. 2008 paper is the wrong reference here as is the Kimura paper.

24) The authors indicate that the AWC transients shown in Biron et al. 2008 are similar to interneuronal imaging (Introduction, fourth paragraph). What data is this assertion based on? Please provide references.

25) The authors should reference the Zimmer paper when discussing the use of nuclear-localized GECIs (subsection “Pan-neuronal calcium imaging coupled with thermal perturbations reveals novel neural functions”, second paragraph) for pan-neuronal imaging.

26) Figure 1 – it would be useful to include in a supplemental what the identities of all imaged neurons are beyond just the few that are labeled. There are clearly neurons that appear to show temperature responses correlated with those in AFD but these are not labeled. There is also little information provided about neuronal identification beyond the description of a few markers that were used. Many of these markers are expressed in multiple cell types. How did the authors unambiguously identify neuronal nuclei? By position as well?

27) Figure 1 – it is important to clearly indicate that the scales on the Y axes are different or replot to place them on the same scale.

28) In the last paragraph of the subsection “Pan-neuronal calcium imaging coupled with thermal perturbations reveals novel neural functions”: The authors appear to be able to detect calcium transients in the AIY soma and refer to Clark et al. 2006 as having showed this before. However, Clark et al. 2006 specifically noted that AIY signals were detected only at a 'varicosity' in the AIY axons and that no signals were detected in the soma.

29) While the authors show a detailed pan-neuronal response map for stimuli in the non-noxious range in Figure 1, why isn't a similar map shown for the nociceptive stimulus which after all is the major topic of the work? This is a pretty strong stimulus – it is important to get an idea of how much of the nervous system responds to this stimulus, and whether there are more L/R asymmetries in the response.

30) Other groups have shown that sensory neuron responses can be driven by other primary responder sensory neurons. Do the authors know whether the AWC responses they observe are due to direct detection of the stimulus or whether AWC responses are being driven by other neurons, for instance AFD or even FLP?

31) Please comment why loss of both AWC neurons results in maintained reversals to 150 mA laser stimulus, but loss of just the AWC(OFF) neuron abolish it?

---

## [Author Response]

[…]

*As you can see in the appended reviews below, all three reviewers value the quality and implication of your work. However, there were some substantial concerns about how this work has been framed and a number of other issues require clarification. None of these requested revisions require additional experimentation. Please address each reviewer’s point in detail. In no order of importance the requested changes are:*

*1) The authors lay claim to a "novel" method for neural-activity screening although the concept is obvious to those in the field and arguably has been "pioneered" – if one can call it that – previously by several other groups. This claim to a novel screening method mars an otherwise great scientific publication. The paper should instead strictly focus on the scientific findings of asymmetric roles for the AWCs in thermal nociception. Specifically, it is recommended that the authors remove the claim of novelty in developing a "pan-neuronal functional screening system" (Introduction, second paragraph) by looking at pan-neuronal calcium responses in response to stimuli. The concept is obvious but, avoiding such discussions, this "technique" was pioneered initially by Kato et al. in their 2015 Cell publication "Global Brain Dynamics Embed the Motor Command Sequence of Caenorhabditis elegans" wherein a pan-neuronal calcium signal is screened during fictive locomotory behavior and the active neurons identified through fluorescent reporters (Figure 1 of the Kato paper).*

*Furthermore, Venkatachalam et al. apply the very same idea to the worm thermosensory circuit in their 2015 publication "Pan-neuronal imaging in roaming Caenorhabditis elegans" uncovering 2 unnamed neurons labeled 62 and 18 (Figure supplement 3 of the Venkatachalam paper). A citation which is strikingly missing from the submitted paper despite being published in PNAS within the same issue as the Nguyen paper which Kotera et al. do cite. Further to this, Kotera and colleagues should cite the work of the Samuel's lab as it is a direct precedent to their own work and sets the stage for their newly published discoveries.*

*Altogether, the claim of a novel technique is a distraction from what is otherwise a great scientific paper.*

Thanks for the suggestions. We have carefully modified the text so that it describes our system as extension of the previously reported systems (Introduction, second paragraph). We also included the citations you have suggested.

*2) The authors use several fluorophores that present odd choices given the availability of newer and better alternatives. This is not problematic but readers will be left to wonder why such choices were made. We recommend the authors address this upfront to direct readers to the alternatives.*

*Chief among these strange choices is GECO1.1. While an improvement on GCaMP3, GECO1.1 fairs poorly when compared to GCaMP6 (the new standard in the field) – please see both the original papers from Zhao et al. 2011 on GECO1.1, Chen et al. 2013 on GCaMP6, and the Neurophotonics comparison in 2015. Presumably the choice was influenced by when Kotera and colleagues began their work. Still, given the array of choices, readers should be advised of best practices.*

*Second, the Discosoma-derived proteins such as DsRed, DsRed2, and mCherry have long been known to have ill effects in worm and, recent yet to be published work by Monica Driscoll's lab show that neurons react poorly to these fluorescent proteins. For this reason, TagRFP is often used in place of DsRed.*

*Third, mNeptune has been superseded by mNeptune2.5 which provides a nearly 2-fold improvement in brightness – published in 2014 by the Lin lab at Stanford.*

*As stated, these are not major issues. The authors should simply address choices for best practices since readers have easy access to strains and plasmids that represent newer better alternatives to the ones used.*

We respectfully disagree with the notion that G-GECO1.1 is a “strange choice”. The original paper describing GECO series have been cited over 300 times and they have been successfully used in many labs around the globe. Both the GECO and GCaMP6 variants display excellent performance in terms of S/N ratio, brightness, stability, kinetics, and dynamic range. Often the truth is one indicator may perform better than the other in some ways in some situations but not in all ways in all the situations. As far as we know there has not been any direct comparison of these indicators in the neurons of *C. elegans*. The Neurophotonics paper you suggested compares these indicators in the mammalian neurons using 2-photon system and evoked action potentials, which is entirely different from what we have measured in this study. Nonetheless, given the popularity of GCaMP6 variants, we mentioned the possibility of improvement using such indicator in the Discussion section (seventh paragraph).

As for the Discosoma-derived FPs, not all variants show phototoxicity in *C. elegans*. DsRed2 was developed to mitigate the slightly toxic effect sometimes observed in the original DsRed, which was associated with protein aggregation. As far as we can tell DsRed2 does not aggregate in *C. elegans* neurons and we have not noticed any toxic effect by this FP. The other reason we chose DsRed2 is its secondary absorption peak at around 488nm which matches the absorption peak of G-GECO. We cannot address “best practice” in this setup because we have not done comparison to TagRFP or any other FPs in this context. Similar argument goes to the mNeptune variants as well.

*3) The authors discover several unreported neurons to be thermo-sensitive. They choose to focus on AWC, a neuron with multiple but conflicting publications stating its thermo-sensitivity. The readers are left to wonder why the authors forewent the more obvious choice of exploring the novel finding of unreported thermo-responsive neurons. The authors should address this choice. Perhaps another paper is forthcoming with their results for RIS. RMDV and SMDV activity (motor neurons innervating the head) are likely reflections of downstream head actions in response to thermal stimulus. This a simple loose end that can receive a quick mention.*

Thank you for the comment. The short answer to the question is that we decided to focus on the noxious stimulus because it has been less studied and connects better with other ongoing projects in the lab. While RIS, RMDV, and SMDV responded to the thermal ramp stimulation, they did not respond to the noxious thermal stimuli. We added this notion to the text (subsection “AWC neurons respond asymmetrically to noxious thermal stimuli”, second paragraph).

*4) The authors have developed an impressive imaging rig capable of delivering thermal stimulus. Presumably a future paper will cover this novel rig but for now, in the spirit of openness, the software should be released to the community for public use in an open source repository and not simply made available upon request. This also benefits the science by providing transparency in the algorithms used to identify neurons, resolve their activity, and the analysis used to assay neural activity and behavioral correlates.*

The software package we developed integrates CUDA-based libraries provided by Dr. Butte, which at time were distributed freely but since then he changed his policy and now they are proprietary. We can probably work on individual requests for scientific projects but we cannot make it open source as is. We do understand the importance of open source spirits, and we would like to rewrite the libraries from scratch in the near future so that the whole package will be freely available. But unfortunately such project is currently beyond our capabilities and cannot accompany this publication.

*5) The supplement should match up the reporters used in the paper with the neurons they were used to identify. As has been the case multiple times in our field, future papers may find that some of the identifications were erroneous. This will help in quick corrections to the knowledge base.*

We have added a supplementary figure (Figure 1—figure supplement 3) to show how we identify the head neurons using reference markers such as glr-1p::mNeptune.

*6) G/R ratio should be explicitly defined in Figure 1, where it is first used, as opposed to Figure 2.*

Thank you. The figure legends are fixed accordingly.

*7) In Figure 1—figure supplement 2, the thermal stimulus should be marked on the individual graphs so as to make sense of the neural traces.*

These experiments follow the same conditions as Figure 1—figure supplement 1, and the thermal ramp stimulus is shown there. We added description explaining the stimulus in the figure legend.

*8) Figure 2—figure supplement 1 shows the poor S/N in AWC-OFF due to the G-GECO measurement against the srsx-3p::GFP reporter. The S/N vastly improves in Figure 2. How was this accomplished? Did the authors interpolate AWC OFF and ON from their thermal responses and then measure their activity with no OFF/ON identification reporters present?*

Yes, that is the case. After we confirmed ON/OFF identities of the calcium transients with srsx-3p::GFP and str-2p::DsRed reporters, along with the calcium transients in nsy-1 and nsy-7 mutants, we were very confident that the direction of calcium transients alone was sufficient to call ON/OFF identities in AWC neurons.

*9) The AWC fates in nsy-1 and nsy-7 mutant background requires a citation (subsection “nsy-1 and nsy-7 mutations alter the functional asymmetry in AWC neurons during noxious thermal stimulation”, first paragraph).*

We have added appropriate citations for nsy-1 and nsy-7 mutants in regard to AWC cell fates.

*10) The term "turn" (first mentioned in the second paragraph of the subsection “nsy-1 and nsy-7 mutations alter the functional asymmetry in AWC neurons during noxious thermal stimulation” and Figure 4) can have multiple behavioral definitions for worms. The authors should be explicit as to what they mean by "turn". Are these omega turns, short reversals that include turning behavior, simple left/right turns?*

The turn in this context is explicitly omega turn. We have added description in the text and figure legend.

*11) The type of statistical tests should be explicitly stated and justified alongside the data – not just in the Methods – so that readers can assess the implicit assumptions made when comparing measured distributions. Furthermore, claims of normal distributions should be backed by histograms or similar representations of the sampling. Worm behavior often deviates from normality and therefore non-parametric tests are often a more appropriate choice. This problem occurs in Figure 4, Figure 4—figure supplement 3, and Figure 5. The authors can switch to violin plots or similar such statistical representations to assure the readers that the sampling is indeed "normal".*

Thanks for the suggestion. We have added description of statistical methods used in the figure legends.

As the reviewer suggests, the behavioral data we collected contained some non-normal distribution: we now use a nonparametric test (Mann-Whitney) to determine if the means are significantly different.

As for the plots, we re-calculated the 83.4% confidence intervals by bootstrapping method in order to more accurately express the CIs of the non-normal distributions. We have changed the figure legends and Methods section accordingly.

*12) There is no N provided for Figure 4. The sample size should be explicitly stated. Furthermore, reversal rate and pausing (Figure 4—figure supplement 1) appear to be different in a nsy-7 mutant background. Did the authors repeat the experiments with larger sample sizes to rule out a role for AWC-ON in reversal rate and pausing as a response to noxious thermal stimulus? The behavioral transition graphs in Figure 4—figure supplement 3 show a clear non-wildtype role for nsy-7 and, inferentially AWC-ON, in response to noxious heat. Yet, these findings receive only a minor mention in the Discussion. We would like some mention at the location where the data is shown as well.*

We added sample sizes for Figure 4. For other figure panels in Figure 4. there are too many data points (over 100) to be listed in the figure legend: we provide minimum number of worms for each strain in the figure legend.

Figure 4—figure supplement 1 shows fraction of behaviors, not the rate of behaviors. The experiment was designed to efficiently screen for behavioral differences among the mutants. Although we used fairly large sample sizes (>100), fractions are just an averaged result of different transitions, and we could not find definitive hypothesis for the roles of AWC-ON neurons in these assays. Instead we focused on AWC-OFF neuron for detailed transition analysis because the difference between N2 and nsy-1 looked promising (p < 0.001). We also added statement in the legend to refer to statistical data in supplementary files.

*13) Claims of behavioral adaptation (and lack thereof) to repeated thermal stimulus in Figure 4 should be backed by a goodness of fit regression to a linear or exponential model of adaptation – or a similar statistic. Scientific claims require stronger evidence than that presented.*

We understand the concern here but believe that our qualitative report of adaptation is clear given the presentation of CI. We are not making claims that the adaptation follows a certain functional model or that responses fall to some quantified ratio. We slightly changed the wording in the text to state that modeling adaptation it itself is not the critical part of this report.

*14) On the figures, the error bars are labeled as 83.4% confidence level but presumably the CI is 95% and 83.4% is termed the corresponding "capture percentage". The correct term should be used.*

As stated in the legends, we calculated error bars to indicate 83.4% confidence intervals. A pair of non-overlapping 83.4% CIs are good visual representation of statistical significance, but not the 95% CIs. We did not use capture percentage in this study.

*15) Figure 5 fails to show not only the N for the variety of conditions tested but, also, the WT response to 150mA and 250mA laser stimulation. This leads to questions as to how AWC ON/OFF ablation was controlled in the statistical analysis of 150mA and 250mA laser stimulation. The authors should address this by showing the missing data and explaining how the statistical tests were performed.*

We added sample sizes for these assays. As for the statistical analysis, all the mutants and cell ablated worms were compared to N2 at 50 mA. Because the mutants and cell ablated worms have different thresholds for the reversal behavior after the stimulation, we believe it is more appropriate to compare them against the wild type that went over the threshold.

*16) The term "sedated" (Discussion, seventh paragraph) is inappropriate for levamisole-mediated paralysis. The worm's neurons are obviously still functional after application of levamisole. We suggest using the term "paralyzed" in place of "sedated".*

Thank you for the correction. We have changed the wording accordingly.

*17) The central claim that AWC_OFF was identified by whole brain recording is powerful, but the only data that shows whole brain recording in Figure 1 is in a regime where AWC_OFF is quiet. It would be helpful to show the whole brain recording data that actually uncovered AWC.*

The whole brain recording in the thermal nociceptive assay had only a handful neurons with strong correlation to the stimuli. In order to show the effectiveness of the whole brain recording, we believe that the results from the thermal gradient assay is more suited because it displays wide range of responses from various neurons, and the thermal gradient stimuli has been used in many prior publications.

*18) An impressive number of cells was identified. They mention using glr-1 expression patterns to help with cell identification, but a more detailed explanation of how they came to their cell identities would be useful.*

We have added a figure (Figure 1—figure supplement 3) that shows how we typically label thermosensory neurons by assessing the relative location to the known neural markers such as glr-1p.

*19) In the text, they claim to have stable recordings for 60 minutes with this technique. This is an impressive claim, and should be supported by data.*

We have added a figure that shows calcium transients of AFD neurons up to 45 minutes. Publication-ready 60-minute recording was not immediately available, so we changed the notion in the text accordingly.

*20) Figure 5 has a typo. The data that correspond to the ablation experiments should be labeled as such.*

We could not locate the typo, but we did add a description that clarifies the meaning of AWC numbers in the legend.

*21) The manuscript was overall quite poorly written which made it difficult to read. The rationalization about different 'hierarchies' in investigating mechanistic underpinnings of behavior was very unclear. After all, calcium imaging is hardly the best or most direct readout of neuronal activity. This was an issue in the Introduction but even more so in the Discussion. This is not the first paper in C. elegans or in any other system to infer neuronal functions from examining stimulus-evoked neuronal activity, and it is inaccurate to portray it as such and to not mention many previous similar reports (for C. elegans – for instance see work from the de Bono lab, Chalasani lab etc.).*

We have modified the text (Introduction, second paragraph, and Discussion, second paragraph) that our pan-neuronal screening system is extension of previously reported systems. We also added more citations to the previous systems.

*22) The authors also do themselves a disservice by not discussing the previous pan-neuronal imaging papers in more detail. Papers from other labs reporting similar imaging methods should be introduced in more detail. In particular, the paper by Venkatachalam et al. from the Samuel lab which specifically reports pan-neuronal imaging of thermal stimuli in freely moving animals is not referenced at all.*

We added more description of the previous works, along with appropriate citations (Introduction, second paragraph).

*23) Related to the above, please check references throughout. In many cases, references are missing altogether or the wrong references are included. For example, the role of AFD in thermosensation (Introduction, fourth paragraph) was first shown by Mori et al. in 1995. The Biron et al. 2008 paper is the wrong reference here as is the Kimura paper.*

Thank you for pointing out the omission. We have corrected the citation errors.

*24) The authors indicate that the AWC transients shown in Biron et al. 2008 are similar to interneuronal imaging (Introduction, fourth paragraph). What data is this assertion based on? Please provide references.*

It is a qualitative comparison but we felt our interneuronal signals from AIY (Figure 2) is similar to Figure 1 in Biron et al. 2008. They are both short pulses with stochastic characteristics. We added a link to the actual figure (Introduction, fourth paragraph).

*25) The authors should reference the Zimmer paper when discussing the use of nuclear-localized GECIs (subsection “Pan-neuronal calcium imaging coupled with thermal perturbations reveals novel neural functions”, second paragraph) for pan-neuronal imaging.*

Thanks for the suggestion. The citation has been added.

*26) Figure 1 – it would be useful to include in a supplemental what the identities of all imaged neurons are beyond just the few that are labeled. There are clearly neurons that appear to show temperature responses correlated with those in AFD but these are not labeled. There is also little information provided about neuronal identification beyond the description of a few markers that were used. Many of these markers are expressed in multiple cell types. How did the authors unambiguously identify neuronal nuclei? By position as well?*

We have added a supplementary figure (Figure 1—figure supplement 3) to show how we identify the head neurons using reference markers such as glr-1p::mNeptune.

*27) Figure 1 – it is important to clearly indicate that the scales on the Y axes are different or replot to place them on the same scale.*

We added a note in the legend to inform the readers about y-scale.

*28) In the last paragraph of the subsection “Pan-neuronal calcium imaging coupled with thermal perturbations reveals novel neural functions”: The authors appear to be able to detect calcium transients in the AIY soma and refer to Clark et al. 2006 as having showed this before. However, Clark et al. 2006 specifically noted that AIY signals were detected only at a 'varicosity' in the AIY axons and that no signals were detected in the soma.*

In fact, the AIY signal in the nucleus was subtle, as stated in our manuscript. Considering the very strong signals in the dendrites of AIY, we would expect to easily miss signals around the nucleus if the calcium indicator is expressed all throughout the cell.

*29) While the authors show a detailed pan-neuronal response map for stimuli in the non-noxious range in Figure 1, why isn't a similar map shown for the nociceptive stimulus which after all is the major topic of the work? This is a pretty strong stimulus – it is important to get an idea of how much of the nervous system responds to this stimulus, and whether there are more L/R asymmetries in the response.*

Unlike the pan-neuronal temperature ramp assays, the pan-neuronal nociceptive assays displayed only a few neurons which had positive or negative correlation to the stimuli. In order to show the pan-neuronal nature of the screening method, we thought the map with a thermal ramp stimulation (more responding neurons) was more appropriate and connected better with prior work.

*30) Other groups have shown that sensory neuron responses can be driven by other primary responder sensory neurons. Do the authors know whether the AWC responses they observe are due to direct detection of the stimulus or whether AWC responses are being driven by other neurons, for instance AFD or even FLP?*

As far as the connectivity map goes, there is no direct input from AFD or FLP to AWC. Of course this does not exclude the possibility of indirect or remote interference from these neurons: we did not have strong enough doubt to experimentally confirm such claim.

*31) Please comment why loss of both AWC neurons results in maintained reversals to 150 mA laser stimulus, but loss of just the AWC(OFF) neuron abolish it?*

The loss of AWC(OFF) causes the reversal defect, and the defect can be abolished with stronger stimulation. We have not fully investigated the mechanism of this abolishment, but there seems to be a threshold for it. It is possible that AWC(ON) may shift the threshold for the abolishment of the reversal defect, but currently we do not have a comprehensive model to explain both the mutant and ablation experiments. We decided to include it here anyway and we hope to be able understand this result in future work.